# Revealing Physiochemical Factors and Zooplankton Influencing *Microcystis* Bloom Toxicity in a Large-Shallow Lake Using Bayesian Machine Learning

**DOI:** 10.3390/toxins14080530

**Published:** 2022-08-02

**Authors:** Xiaoxiao Wang, Lan Wang, Mingsheng Shang, Lirong Song, Kun Shan

**Affiliations:** 1Key Laboratory of Reservoir Aquatic Environment, Chongqing Institute of Green and Intelligent Technology, Chinese Academy of Sciences, Chongqing 400714, China; wangxiaoxiao@cigit.ac.cn (X.W.); msshang@cigit.ac.cn (M.S.); 2University of Chinese Academy of Sciences, Beijing 100049, China; 3School of Artificial Intelligence, Chongqing University of Education, Chongqing 400147, China; wanglan@cque.edu.cn; 4College of Computer Science and Technology, Chongqing University of Posts and Telecommunications, Chongqing 400065, China; 5State Key Laboratory of Freshwater Ecology and Biotechnology, Institute of Hydrobiology, Chinese Academy of Sciences, Wuhan 430072, China; lrsong@ihb.ac.cn

**Keywords:** *Microcystis* blooms, microcystins, nutrient, zooplankton, machine learning, risk management, Lake Taihu

## Abstract

Toxic cyanobacterial blooms have become a severe global hazard to human and environmental health. Most studies have focused on the relationships between cyanobacterial composition and cyanotoxins production. Yet, little is known about the environmental conditions influencing the hazard of cyanotoxins. Here, we analysed a unique 22 sites dataset comprising monthly observations of water quality, cyanobacterial genera, zooplankton assemblages, and microcystins (MCs) quota and concentrations in a large-shallow lake. Missing values of MCs were imputed using a non-negative latent factor (NLF) analysis, and the results achieved a promising accuracy. Furthermore, we used the Bayesian additive regression tree (BART) to quantify how *Microcystis* bloom toxicity responds to relevant physicochemical characteristics and zooplankton assemblages. As expected, the BART model achieved better performance in *Microcystis* biomass and MCs concentration predictions than some comparative models, including random forest and multiple linear regression. The importance analysis via BART illustrated that the shade index was overall the best predictor of MCs concentrations, implying the predominant effects of light limitations on the MCs content of *Microcystis*. Variables of greatest significance to the toxicity of *Microcystis* also included pH and dissolved inorganic nitrogen. However, total phosphorus was found to be a strong predictor of the biomass of total *Microcystis* and toxic *M. aeruginosa*. Together with the partial dependence plot, results revealed the positive correlations between protozoa and *Microcystis* biomass. In contrast, copepods biomass may regulate the MC quota and concentrations. Overall, our observations arouse universal demands for machine-learning strategies to represent nonlinear relationships between harmful algal blooms and environmental covariates.

## 1. Introduction

The colony-forming genus *Microcystis* is exemplified as the most notorious cyanobacterial bloom former in freshwater systems [1]. Indeed, *Microcystis* possess a range of unique eco-physiological traits that enable them to dominate in nutrient-enriched conditions and evolving climates [2]. Numerous strains can produce hepatotoxic polypeptides known as microcystins (MCs) [3]. Cyanotoxins can cause public health implications via several exposure routes. For instance, skin exposure may lead to rashes, hives, or skin blisters. Swallowing water containing cyanotoxins can result in acute gastroenteritis. Inhaling water droplets from recreational activities can cause allergic reactions such as runny eyes and asthma-like symptoms [4]. Worst of all, exposure to MC-LR (the most toxic MC variant) at sublethal doses may cause continual apoptotic cell death in the liver [5]. MCs can remain in water due to their half-life of days to weeks and also enter the food web by biotransformation and bioaccumulation. The World Health Organization proposed a safety limit of 1 μg/L MC-LR in drinking water and a chronic tolerable daily intake (TDI) of 0.04 μg kg^−1^ body mass per day for human consumption.

Prior studies have explored which environmental factors are essential for the *Microcystis* blooms toxicity to deal with this safety concern [6,7]. *Microcystis* blooms often comprise toxic and non-toxic strains. Thus, most model strategies always focused on establishing the relationship between *Microcystis* morphotypes and associated MCs [8,9], or characterising the dynamics of MCs-producing genotypes and MCs production [10,11]. However, high temporal and spatial variations in MC-producing cell and MC concentrations cannot be directly predicted by toxic *Microcystis* abundance [12]. Complicating the prediction may contribute to the MCs production amongst different strains or even within the same species manipulated by multiple environmental factors [13]. Due to the complex interaction between biotic and abiotic factors, the capacity of mathematic models to simulate MCs produced per cell or MCs released into water column remains limited [14].

In recent years, data-intensive methods have been among the most rigorous approaches for studying complex algal blooms dynamics with field survey data. Various tree-based models have been established for algal parameters forecasting, including the classification and regression tree, random forest, and extreme gradient boosting [15,16]. However, the abovementioned methods did not consider the uncertainty of the explanatory variables. As a probabilistic-based ensemble method, the Bayesian additive regression tree (BART) may offer advantages in regression tasks and allow a quantitative assessment of input uncertainty. As a result, the BART model has shown great potential in predictions on environmental issues, such as hourly streamflow forecasting [17] and daily PM_2.5_ concentrations [18].

Developing MCs models based on field surveys is securing large expansive data set, because some studies only sample potential variables of interest. Therefore, the data set often contains incomplete information with amounts of missing values [19]. Traditional feature selection approaches are invalid when the number of missing values far exceeds the number of measured values. This phenomenon of daunting variables mismatches has impelled scientists to use data imputation to mitigate the impacts of missing values [20]. Many imputation methods derive the conditional distribution of the missing data based on the observed data [21]. Recently, the non-negative latent factor (NLF) analysis has been widely used in predicting missing values from various areas, such as image information extraction and recommendation systems [22,23].

In this paper, we proposed the NLF-BART model, a hybrid machine-learning framework, as a solution to evaluate the drivers of MCs risks based on a data set containing 22 sampling stations. Unlike the traditional method, NLF-BART rely on a probabilistic statistic rather than a deterministic one to describe the relationship among variables. We hypothesized that the risks of MCs could be accurately predicted by combining the selected environmental parameters with the biotic parameters. To do so, missing values of MCs concentrations were firstly imputed using an NLF-based algorithm. Then, we applied the BART method to the assessment of cyanotoxin dynamics. To our knowledge, this is the first investigation to apply the Bayesian ensemble-modelling methodology to assess the risk of harmful algae blooms.

## 2. Results

### 2.1. Summary of Observations

The PCA biplot showed distinct patterns in cyanobacterial community distributions in Lake Taihu (Figure 1a). The first two axes of PCA accounted for 54% (32% plus 22%) of total variations, which were mainly described by the biomass of *Microcystis*, *Anabaena*, and *Microcystis aeruginosa*. When the sampling dates were divided into four periods, toxic *Microcystis aeruginosa* corresponded to the autumn season, and non-toxic genera such as *Microcystis wesenbergii* and *Microcystis novacekii* occupied considerable biomass in the summer. Filamentous taxa *Anabaena* and *Aphanizomenon* were dominant in the winter and early spring. The redundancy analysis (RDA) was further performed to obtain a sketch of the environmental drivers of the *Microcystis* blooms toxicity (Figure 1b). The RDA results demonstrated that the total biomass of *Microcystis* was affected by water temperature (WT), pH, total phosphorus (TP), and protozoa biomass. In contrast, shade index (SI), total nitrogen (TN), and copepods biomass were correlated with MCs quota and concentrations.

### 2.2. Modeling of Intracellular and Extracellular MCs in Lake Taihu

#### 2.2.1. Imputation of Missing MCs Data Using NLF Machine Learning

Missing values of MCs were imputed using the NLF algorithm by using the information on environmental factors, cyanobacterial communities, and zooplankton assemblage. The imputation performance in terms of R^2^ for the intra- and extra-cellular MCs were 0.90 and 0.99, respectively, demonstrating that the NLF model achieves a promising accuracy for imputing the timing and magnitude of MCs in the studied lake (Figure 2). From the monthly dynamics of the measured and imputed MCs, both intra- and extra-cellular MCs exhibited a clear annual seasonality pattern with high values in the summer and autumn (Figure 3). Regarding the spatial distributions, the high hazard of MCs was found north-westward across the whole research site (Appendix A).

#### 2.2.2. Fitting Microcystin by Bayesian Additive Regression Trees

The BART model’s performance on MCs predictions was presented by plotting the parity plot of the predicted and actual median values. As shown in Figure 4, the BART model fits the training data well, as measured by the predictive range (95% prediction interval) for each data point which is about 96.57% and 91.42% for intra- and extra-cellular MCs predictions, respectively. In comparison, the BART model also performed as well on testing data with about 91.28% and 84.38% accuracy for *Microcystis* biomass and MCs concentrations, respectively (Appendix A).

It is essential to check the BART’s assumption of the Gibbs sampler convergence. Appendix A shows the BART model had a stable evolution based on the given parameters after about 800th, 400th, and 200th MCMC iterations for *Microcystis* biomass, intra-, and extra-cellular MCs prediction, respectively. The BART method was compared with two traditional models, namely multiple linear regression (MLR) and random forests (RF). Table 1 shows the two metrics in terms of RMSE and R^2^ of the three models in all predictive tasks, and one can conclude that the prediction accuracy of BART was more effective than the compared models as mentioned above by the Wilcoxon rank-sum test.

### 2.3. Sensitivity of the Toxicity of Cyanobacterial Blooms to Abiotic and Biotic Variables

#### 2.3.1. Variables of Permutation Importance

One of the BART model’s essential advantages is its ability to rank the importance of input variables. The relative importance of an explanatory variable is evaluated by its frequency. It is selected by calculating the splitting rule across all posterior samplers. Figure 5 presents the relative importance of twelve abiotic and biotic variables. Phosphorus concentration and WT were associated with the most significant permutation importance in the *Microcystis* biomass model. pH and cladocerans biomass were the most important in the biomass of *Microcystis aeruginosa*. These variables were also crucial for the prediction of intracellular MCs. Extracelluar MCs concentrations were sensitive to SI, dissolved inorganic nitrogen (DIN), and WT.

#### 2.3.2. Nonlinear Relationships between the Predictors and MCs

The partial dependence plots correspond to the relationships linking response and explanatory variables. The curves are the average modelled values across the range of explanatory variable observations. Non-fixing *Microcystis* biomass (B_M_) displayed a relatively strong partial dependence on WT, wind speed (WS), TP, and protozoa biomass (Figure 6). The curve depicts a threshold in the relationships between WT and B_M_. Specifically, partial dependence plateaued at 0~25 °C but increased sharply when WT exceeded 25 °C; *Microcystis* began to dominate at low wind speed and high total phosphorus.

With regards to MCs, the steepest curves were associated with SI. Partial dependence decreased sharply with the increase of SI, revealing a slight sensitivity of MCs production under light limitation conditions (Figure 7 and Figure 8). MCs began to accumulate at low DIN and higher copepods biomass. In addition, TN and soluble reactive phosphoru (SRP) emerged as explanatory variables on extracellular MCs. At the same time, nutrients play a less role in regulating intracellular MCs. Intra- and extra-cellular MCs exhibited a conversely strong partial dependence on WT and pH. The critical water temperature above which intracellular MCs gradually increased was ~20 °C, in contrast to an ambiguous one for extracellular MCs. We detected that intracellular MCs had a robust positive relationship with pH. However, the highest concentration of extracellular MCs mainly occurred at the pH range of 7.5~8.0.

## 3. Discussion

Until now, the effects of environmental factors on the occurrence and concentrations of cyanotoxin have been well documented in lab or field-scale studies [13]. However, only a few studies eventually proposed a predictive model to estimate MCs production based on different limiting factors [14,19]. In this study, we developed a Bayesian ensemble-modelling methodology to improve the predictions made for the MCs concentrations and incorporate the environmental variables’ uncertainty into the Microcystis blooms management scheme. Bayesian machine-learning approaches can display a ranking of how strongly the inputs drive the output response, and this manner helps understand the uncertainty of the model. Data-driven models are helpful when reliable monitoring data under the different spatial and temporal scales of environmental conditions is available [24]. However, there is often a lack of sufficient MCs data in the training model due to experience-dependent and time-consuming measurements. Machine-learning technologies could extract valuable knowledge from the incomplete dataset to enhance model reliability and applicability [25]. This incomplete data analysis also has become a hot research topic in data science [26].

The predictive performance of BART was substantially based on twelve abiotic and biotic variables, implying that the prediction of the MCs should require knowledge of environmental conditions. Some researchers tried to predict cyanotoxin content via the growth rate of nutrient-limited species [27,28]. Nevertheless, other researchers suggested cyanotoxin production may be a function of several environmental factors, including photosynthetically active radiation, nutrient form concentrations, or water turbulence [29,30,31]. In this study, the importance of the environmental factors was ranked using the partial dependence plots, which provided the key to directly understanding what drives changes in bloom toxicity and toxin yield at the environmental level.

The light conditions, indicated by SI, played the most critical role in the variations of observed MCs. In our dataset, SI was correlated with increased *Microcystis* biomass but was negatively correlated with MCs quota and concentrations. This finding is consistent with previous experimental results that non-toxic strains of *Microcystis* are better competitors in the presence of light than toxic strains under low-light conditions [32,33]. Since most sites in our studies are prone to light-limitation, positive effect of light availability on microcystin production up to the point where the maximum growth rate is reached was in line with the findings by Wiedner [29]. Interestingly, our results suggested that pH was positively associated with MCs produced per cell, while it was negatively related to MCs in water columns. This was in line with the previous evidence that the MCs production increased at more alkaline pH conditions because MCs-producing species would outcompete other phytoplankton [34,35].

Nutrient control remains the most sustainable manner available to mitigate the toxic cyanobacteria blooms. The trends from partial dependence plots reinforced that MCs as nitrogen-rich metabolites are reduced disproportionately under nitrogen limitation [14]. Importance analysis showed that DIN was a better indicator than TN in MCs prediction because the luxury uptake of nitrogen is limited. Total nitrogen contains a significant component of non-available organic nitrogen in hypereutrophic lakes. Meanwhile, decreased DIN concentrations led to low biomass of *M. aeruginosa* instead of total *Microcystis* biomass, thus supporting findings that non-toxic strains of *Microcystis* outlive toxic strains when a small number of nutrients are available [36]. Conversely, the relationship between TP and MCs is prone to unimodal models rather than linear responses. *Microcystis* has a competitive advantage in low concentrations of phosphorus because colonial *Microcystis* has strong buoyancy control via vertical migration between the upper and lower parts of the water column, thereby allowing them to pool phosphoric sediments even in oxygenated conditions [37]. In comparison, MCs were much higher at the increased SRP, because sediment P release may increase nitrate concentration in the ecosystem [38].

Another biotic factor influencing the dynamics of blooms toxicity is the role of zooplankton grazing pressure. From BART results, metazoan zooplankton, including cladocerans and copepods, were positively correlated with the increase in MC cell quotas and dissolved MCs. The implications of top-down controls could interpret these relationships. Cyanobacteria blooms and their dead organisms were confirmed as potential food sources for the secondary production of lakes [39,40]. During the *Microcystis* blooms in Taihu Lake, large-sized *Microcystis* (over 100 μm) taking up more than 80% of total biomass could account for the potentially high MCs concentration [41]. However, non-living organic matter of blooms, heterotrophic bacteria, and protozoa form a microbial loop, supporting a bulged metazoan zooplankton biomass. In addition, small-bodied metazoan zooplankton has been reported to develop some tolerance to toxic cyanobacteria [42], because they were capable of consuming filamentous or less toxic strains [43].

Although the underlying mechanism of MCs production is unclear, results from our building model corroborate previously experimental findings, which indicate that the hazardous of *Microcystis* toxicity increases when temperature increases, light is available, and dissolved nitrogen is depleted. The interactions between zooplankton and cyanotoxins can also provide important information that allelopathic controls compounds did not significantly influence the reduction of zooplankton biomass. However, several open issues about predicting MCs using a machine learning approach should be considered. For example, the use of some low-cost variables might improve the prediction performance of MCs prediction [9]. We will plan to test continuously meteorological and hydrologic factors that can be used to improve our proposed model’s performance. Furthermore, many machine learning approaches have been recently used to identify correlations and assess the drivers of complex ecological dynamics. A future scientific challenge is to determine whether there is any causal link among environmental factors, toxic strains and cyanotoxins.

## 4. Conclusions

The NLF-BART represents nonlinear relationships between MCs and environmental covariates in Lake Taihu. The promising imputation performance demonstrates that the NLF model can effectively impute missing values of MCs based on their latent patterns. The BART model provided a reference for the future prediction of MCs by identifying environmental variables most likely to be useful. Model results exemplified that the hazardous of *Microcystis* toxicity may be promoted when temperature increases, light is available, and dissolved nitrogen is depleted. Metazoan zooplankton was positively correlated with the increase in MC cell quotas, which may be better to understand the potential effects of allelopathic controls on zooplankton. Overall, this study is a case to guide water managers or researchers to utilise machine-learning approaches to study harmful algal blooms.

## 5. Material and Methods

### 5.1. Site and Data Description

Lake Taihu is the third largest lake in China (2338 km^2^) and has a mean depth of only 2.7 m. Cyanobacterial blooms have been observed since the 1980s. *Microcystis* began to dominate in the cyanobacterial community in the spring and summer. In this study, we analysed a monthly monitoring dataset from 22 sampling sites between October 2008 and October 2010, which contains monthly observations of water quality, cyanobacterial taxa, and zooplankton assemblage.

During each transect, water samples were collected at 0.5 m depth of surface layer. Physiochemical factors used in the building model, including water temperature (WT, in °C), Secchi Disk depth (SD, in cm), pH, wind speed (WS, in m/s), total nitrogen (TN, in mg/L), total phosphorus (TP in mg/L), dissolved inorganic nitrogen (DIN, in mg/L), and soluble reactive phosphorus (SRP, in mg/L). More detailed data information can be described by Wu et al. (2016) [44] and Shan et al. (2019) [45]. Shade index (SI, in m/m) was defined as the ratio of lake depth to SD according to Kosten et al. (2012) [46].

One liter of water sample was collected with a polymethyl methacrylate sampler and preserved with acid Lugol’s iodine solution (1% final conc.) for the identification of phytoplankton assemblages. Phytoplankton species were identified according to commonly used phytoplankton monographs and counted three times with a Sedgwick-Rafter counting chamber under Olympus CX31 optical microscope. The *Microcystis* morphospecies were classified according to the descriptions from Komárek and Komárková (2002) [47]. Algal cell counting was estimated by picking each *Microcystis* morphotype colonies from fresh samples, fixing each with 0.1 mL of the Lugol’s iodine solution and shaking it at 120 rpm until the colonies became unicellular. The average number of cells in each type of morphospecies was further multiplied to the number of colonies present to estimate the abundance of each morphospecies. *Microcystis* biomass (B_M_) was calculated on the basis of abundance (Biomass = algal density (1 g/cm^3^) × size × abundance).

Quantitative samples for protozoans and rotifers were prepared and preserved using the procedures used for the phytoplankton. Quantitative samples (20 L) for copepods and cladocerans were filtered through a 69 µm net, backwashed into a bottle with filtered lake water, and preserved in 4% formalin solution. The samples for protozoa analysis used a volume of 0.1 mL in with 20 mm × 20 mm settling chambers. Samples for rotifer, copepod, and cladoceran analysis used a volume of 1.0 mL with 50 mm × 20 mm × 1 mm settling chambers. Zooplankton biomass was calculated as wet weight per individual × abundance.

Enzyme-linked immunosorbent assays measured extracelluar microcystins (dMCs, in μg/L) concentrations between October 2008 and September 2009. Water samples (100 mL) were filtered through a GF/C filter to remove plankton cells. The filtrates were measured by 96 wells filled with MCs for enzyme-linked immunosorbent assays. The samples were analysed in triplicate and compared with a 0.1 μg/L to 2.0 μg/L calibration curve of MC-LR standard (provided by the Institute of Hydrobiology, Chinese Academy of Sciences) performed on each plate. Enzyme reactions were initiated by adding a substrate solution (0.1 M sodium acetate buffer with pH of 5.0) that contained 100 μg/mL of TMBZ and 0.005% (*v*/*v*) H_2_O_2_ and stopped with 1 M H_2_SO_4_. Absorbance at 450 nm was measured with a microtiter plate reader.

Intracelluar microcystins (cMCs, mg/g dry weight) were extracted with 90% (*v*/*v*) aqueous methanol, and extracts have seeped through Sep Pak C18 cartridges. Finally, cell-bound MCs were eluted in solutions with 1 mL 50% (*v*/*v*) chromatographic pure methanol (Thermo Fisher Scientific, Waltham, MA, USA) and stored at −20 °C for HPLC analysis. The details of MCs measurements can be found in previous studies by Wu et al. (2014) [8] and Hu et al. (2016) [48].

### 5.2. Imputation of Missing MCs Using Non-Negative Latent Factor

The non-negative latent factor (NLF) method was applied to impute missing MCs quota and concentration between October 2009 and October 2010, based on the complete information of environmental factors, cyanobacterial species, and zooplankton communities. Thus, more than fifty percent of intra- or extra-cellular MCs concentrations in the monthly monitoring dataset were imputed using NLF algorithm. The NLF model has been proposed to extract useful knowledge from large-scale incomplete data sets filled with non-negative data [49], and schematic diagram of the NLF method is shown as follows (Figure 9):

Given an incomplete matrix of multivariate data *M* with missing values, *M* is |T|×|V| matrix where each element $m_{t,v}$ is multivariate data and Λ and Γ are known and unknown entry sets of *M*. LF model tries to build a low-rank approximation to an incomplete multivariate matrix. Given *M* and Λ, an NLF model builds *M*’s rank-*f* approximation $\hat{M}=PQ$, where *P* is |T|×|f|, *Q* is |f|×|V|. Note that *f* is the dimension of the LF space. NLF carries out this approximation process subject to the non-negative constraint, i.e., P,Q≥0. Details associated with the NLF method are provided in Luo et al. (2017) [22].

### 5.3. Identification of the Driving Factors for MCs Risks

Principal components analysis (PCA) and redundancy analysis (RDA) were carried out to investigate the relationship among the environmental factors, cyanobacteria communities, and microcystin concentrations. All variables were log-transformed (X_i_ + 1) to ensure that the residuals from the different statistical analyses were normally distributed and homogeneous. This analysis was performed using the “vegan” package in R. With regard to the spatial distributions, the inverse distance weight as interpolation method was used to visualize the distribution of intra- or extra-cellular MCs in Lake Taihu.

We utilised the Bayesian additive regression tree (BART) to quantify and characterise the responses of *Microcystis* biomass and MCs concentrations to environmental drivers. Bayesian additive regression tree is a machine learning algorithm developed by Chipman [50]. The BART was selected for our study because it allows a quantitative assessment of predictor uncertainty [51]. Its ability to estimate flexibly nonlinear problems and nonparametric functions is well suited to many modern statistical regression problems.

Based on BART results, the permutation importance and partial dependence are conducted to quantify how *Microcystis* bloom toxicity responds to relevant physicochemical characteristics and zooplankton assemblages. Permutation significance refers to the change in mean squared error (MSE) when a fitted model is run with a randomly permuted explanatory variable [52]. Partial dependence is a measure of the effect of an explanatory variable on the response variable over the observed range of each explanatory variable [53]. If the partial dependency fluctuates significantly within a narrow range of explanatory variables, it means that the response variables have a high relative sensitivity to input variables. BART model was performed in the “bartMachine” package of R [54].

### 5.4. Model Comparison and Evaluation

Two common algorithms, namely multiple linear regression (MLR) and random forests (RF), were applied to analyse the same data to evaluate the effectiveness of the selected BART model in MCs prediction. MLR is a fundamental statistical method for learning the predictive relation between several explanatory variables and a response variable [54]. RF is a traditional tree-based method which consists of an ensemble of randomised classification and regression trees [55]. RF computation was performed in the “RandomForest” package in R [56].

This research randomised and partitioned the dataset into five equally sized sets based on a discretised dataset. Five repetitions were performed using a training set (80% of the data) and a testing set (20% of the data). The training sets were used to learn the model. This study utilised two performance metrics, namely *RMSE* and *R^2^*.

These statistics are calculated using Equations (1) and (2), respectively:(1)$$RMSE=\sqrt{\frac{1}{n}\sum_{i=1}^{n}{\left(P_{i}-O_{i}\right)}^{2}}$$
(2)$$R^{2}=\frac{\sum_{i=1}^{n}{\left(P_{i}-{\bar{O}}_{i}\right)}^{2}}{\sum_{i=1}^{n}{\left(O_{i}-{\bar{O}}_{i}\right)}^{2}}$$
where $P_{i}$ and $O_{i}$ are the predicted and observed values.

## Figures and Tables

**Figure 1 toxins-14-00530-f001:**
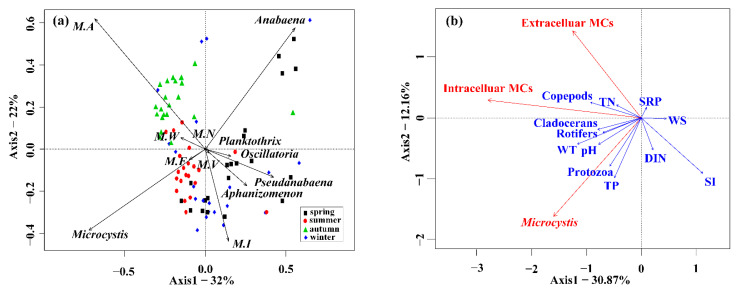
(**a**) Principal component analysis illustrating associations among cyanobacterial communities; (**b**) Redundancy analysis illustrating relationships between environmental factors and *Microcystis* blooms toxicity. Cyanobacteria species are in *italics*. *Microcystis aeruginosa* = M.A; *Microcystis wesenbergii* = M.W; *Microcystis novacekii =* M.N; *Microcystis ichthyoblabe* = M.I; *Microcystis flos-aquae* = M.F; *Microcystis viridis =* M.V.

**Figure 2 toxins-14-00530-f002:**
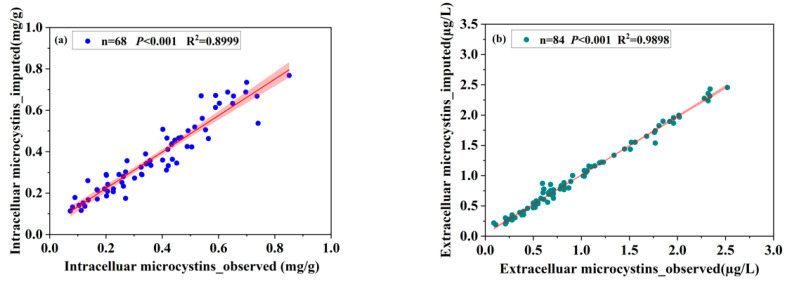
(**a**) Relation between the observed and the imputed intracellular MCs from the NMF model. (**b**) Relation between the observed and the imputed intracellular MCs from the NMF model. *P* was the value of the statistical correlation and obtained by test of significance.

**Figure 3 toxins-14-00530-f003:**
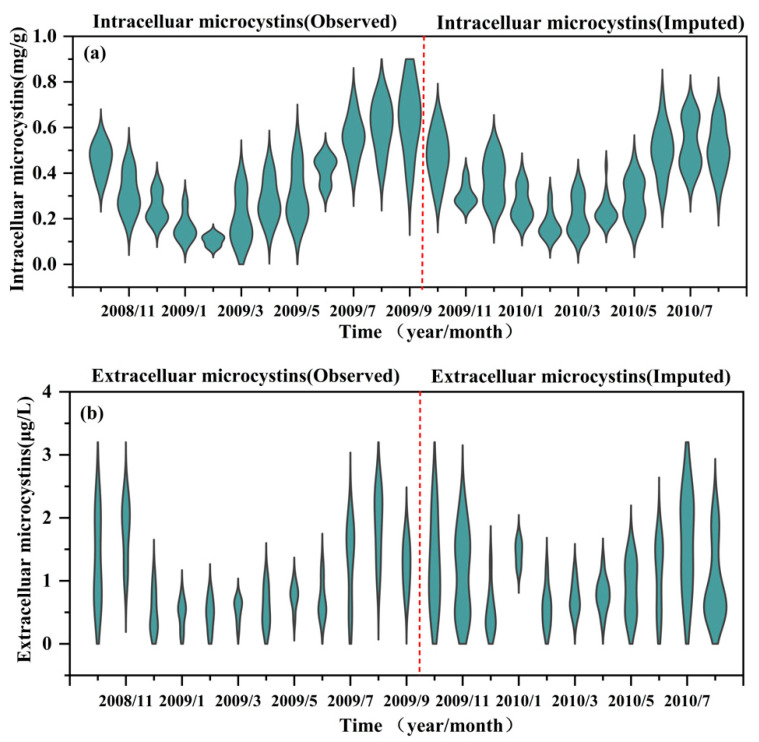
(**a**) The compared monthly dynamics of intracellular MCs based on observed (2008/10~2009/09) and imputed results (2009/10~2010/08). (**b**) The compared monthly dynamics of extracellular MCs based on observed (2008/10~2009/09) and imputed results (2009/10~2010/08).

**Figure 4 toxins-14-00530-f004:**
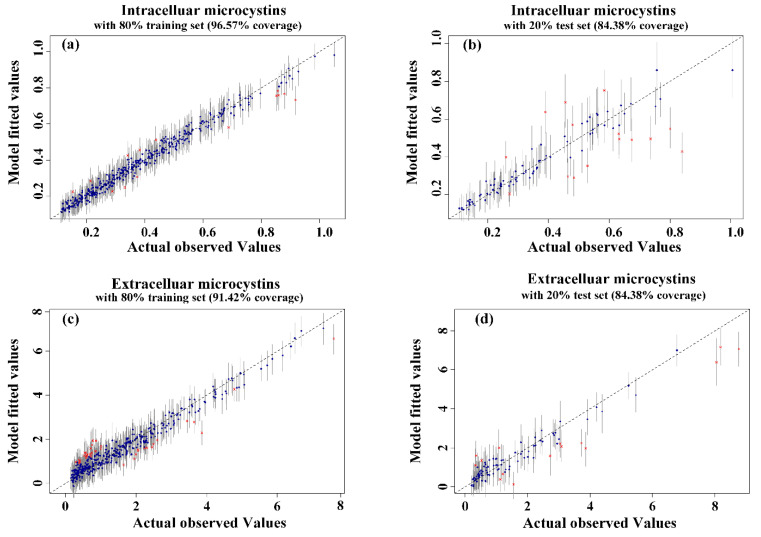
Accuracy validation of the BART model for intracellular MCs with 95% confidence interval for training dataset (**a**) and test dataset (**b**). Accuracy validation of the BART model for extracellular MCs with 95% confidence interval for training dataset (**c**) and test dataset (**d**). It was performed by using 80% of the data as a training set and 20% of the data as a testing data.

**Figure 5 toxins-14-00530-f005:**
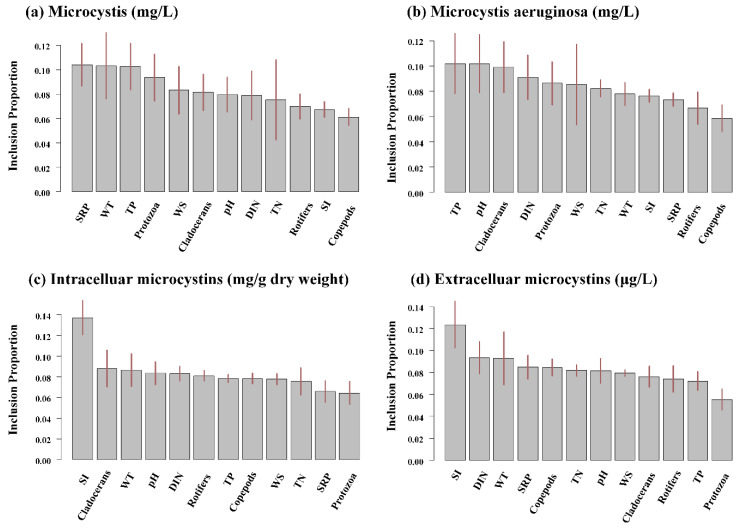
Variable importance (measured with inclusion proportions—essentially the proportion of times that a variable has been used in a splitting rule) of each environmental variable for predicting total *Microcystis* biomass (**a**), *Microcystis aeruginosa* biomass (**b**), intracellular MCs (**c**), and extracellular MC (**d**) in the BART models. The ranges at the top of the bars represent the 95% confidence intervals of the predictors.

**Figure 6 toxins-14-00530-f006:**
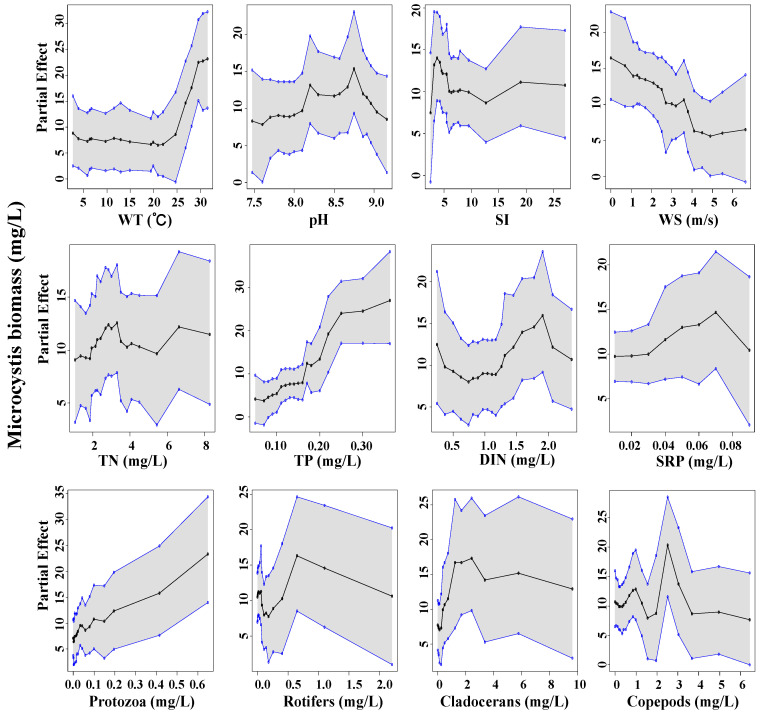
Partial dependence plots for the responses of *Microcystis* biomass (*y*-axes) to the predictors (*x*-axes). Each panel reflects the densities of the explanatory covariate’s observed values. Solid black line equals the median partial dependences from BART model, and corresponding shaded areas are bound by the min and max partial dependences.

**Figure 7 toxins-14-00530-f007:**
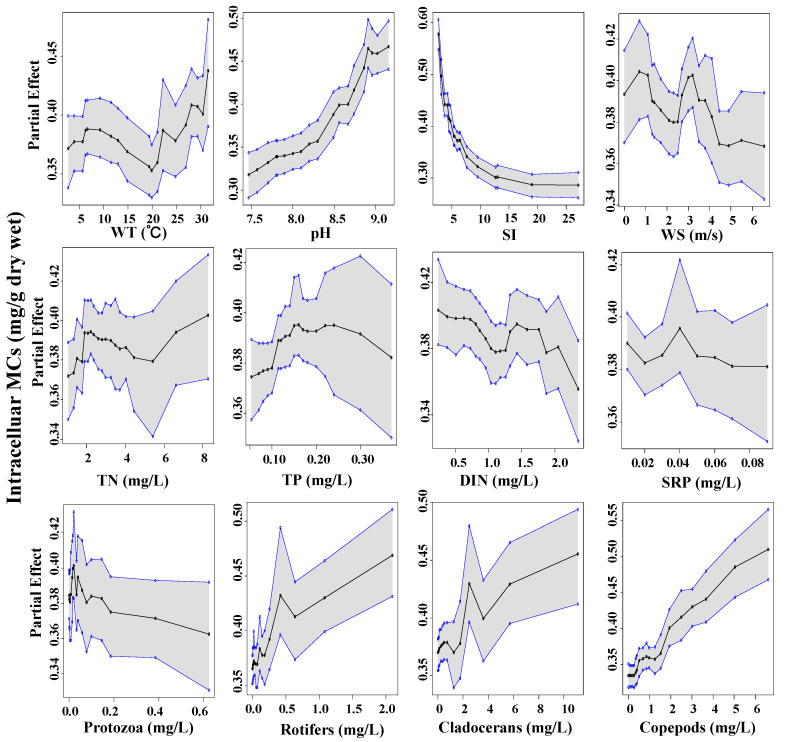
Partial dependence plots for the responses of intracellular MCs concentrations (*y*-axes) to the predictors (*x*-axes). Each panel reflects the densities of the explanatory covariate’s observed values. Solid black line equals the median partial dependences from BART model, and corresponding shaded areas are bound by the min and max partial dependences.

**Figure 8 toxins-14-00530-f008:**
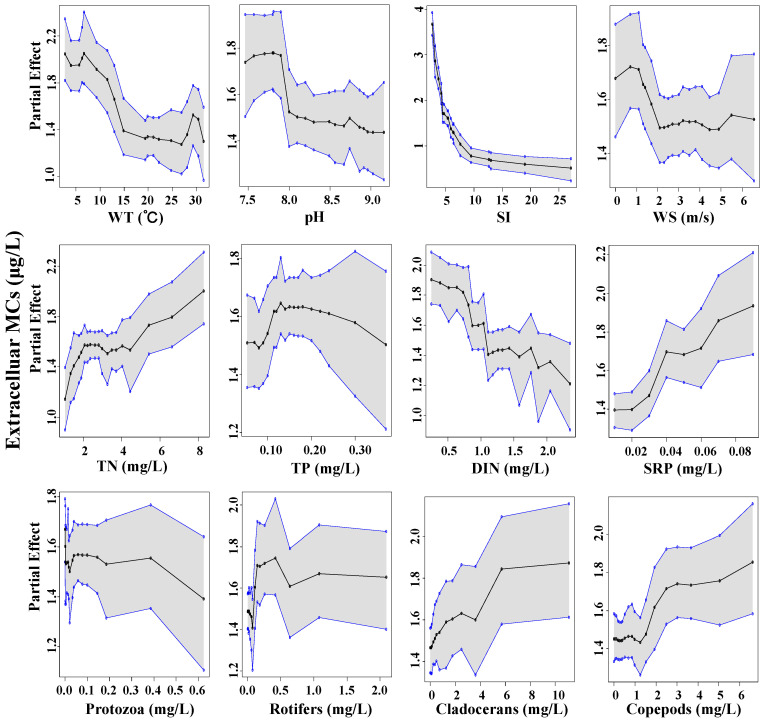
Partial dependence plots for the responses of extracellular MCs concentrations (*y*-axes) to the predictors (*x*-axes). Each panel reflects the densities of the explanatory covariate’s observed values. Solid black line equals the median partial dependences from BART model, and corresponding shaded areas are bound by the min and max partial dependences.

**Figure 9 toxins-14-00530-f009:**
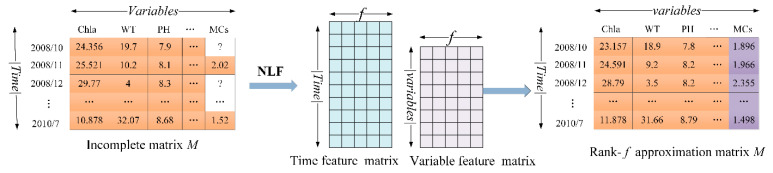
Schematic diagram of structure of NLF model.

**Table 1 toxins-14-00530-t001:** Comparative results of RMSE and R^2^ among different predictive tasks for monthly *Microcystis* biomass, intra-, and extra-cellular MCs. Three models included random forests (RF), multiple linear regression (MLR), and Bayesian additive regression trees (BART).

	Metric	RF	MLR	BART
** *Microcystis* ** **biomass**	RMSE	39.320 ± 0.109	29.854 ± 7.356	**29.924 ± 1.475**
	R^2^	0.561 ± 0.004	0.336 ± 0.033	**0.688 ± 0.036**
**Intracellular** **MCs**	RMSE	0.093 ± 0.001	0.115 ± 0.009	**0.088 ± 0.001**
	R^2^	0.793 ± 0.004	0.646 ± 0.030	**0.807 ± 0.006**
**Extracellular** **MCs**	RMSE	0.899 ± 0.008	1.116 ± 0.180	**0.584 ± 0.020**
	R^2^	0.865 ± 0.003	0.376 ± 0.048	**0.902 ± 0.007**
**Statistical Analysis**	*p*-value	0.125	0.375	---
	F-rank	2.3333	2.3333	**1.3333**

## Data Availability

The data presented in this study are available in this article and Appendix A.

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
