# Peer review of "Revealing Physiochemical Factors and Zooplankton Influencing Microcystis Bloom Toxicity in a Large-Shallow Lake Using Bayesian Machine Learning"

_toxins, 2022, doi:10.3390/toxins14080530_

Round 1

Reviewer 1 Report

Dear Editor,

I recommend that the manuscript be accepted for publication with major revision if the indicated changes are made and the material and methods section is complete adequately. The manuscript is well written, the narrative is very fluid with a good level of English. The discussion is consistent with the results obtained and the specialized literature on the subject. The results presented are of general and applied interest, are coherent and are generally well argued. The use of valid statistical methods that allow inferring the fundamental parameters in the proliferation of cyanobacteria and the concentration of cyanotoxins. In addition, the growing and recurring impact of toxic cyanobacterial blooms is currently a highly relevant issue due to the potential risk that it may entail for human health and the lack of mitigation tools.

The statistical methods proposed are adequate to know nonlinear relationships between cyanobacetria blooms and environmental covariates. However, the methodological sections are incomplete, and some procedures and results remain unclear.

The main deficiency is to understand how the concentrations of Microcystis spp., and the different groups of zooplankton, have been measured. These results are essential to establish the statistical model, so they must be clarified.

Major questions:

There are important omissions in section 4 `material and methods´ (see below) about the results and analyzes performed:

(i)The methods of measurement and/or analysis of physico-chemical parameters and biological assemblages are not specified (lines 270-276). I understand that the physico-chemical parameters measures are described in Shan et al. (2019) but it is not clear. This sentence (lines 269-270: `More detailed data information can be described in 269 Shan et al. (2019) [45].´ should move to the line 276 at the end of the physico-chemical parameters description.  Or include the reference to Wu et al., 2016 (Y. Wu, L. Li, L. Zheng, G. Dai, H. Ma, K. Shan, H.D. Wu, Q. Zhou, L. Song. Patterns of succession between bloom-forming cyanobacteria Aphanizomenon flos-aquae and Microcystis and related environmental factors in large, shallow Dianchi Lake, China. Hydrobiologia, 765 (1) (2016), pp. 1-13) in the line 276 (see Shan et al., 2019).  The authors may also cite both previous works, Wu et a. 2016 and Shan et al. 2019, since it is sometimes difficult to follow the cites through intermediate references.

Figure 1b (line 90): RDA analysis show in the graph the parameter SRP (may be… = SRP) but in the section 4.1 `Site and description´ the authors cited (line 273) `dissolved inorganic phosphorus (DIP, in mg/L)´, are they the same parameter?... If there are no chemical differences between SRP and DIP?? is DIP included in the statistical analyses?... should correct the name of this parameter in Figure 1b or specify this parameter in the description of section 4.1

(ii)The description about quantitative analysis of biological assemblages (i.e., Microcystis spp., rotifers, cladocerans, copepods, protozoa) not directly detailed in Shan et al., 2019. This reference refers to Hu et al., 2016 (L. Hu, K. Shan, L. Lin, W. Shen, L. Huang, N. Gan, L. Song Multi-year assessment of toxic genotypes and microcystin concentration in northern lake Taihu, China. Toxins, 8 (1) (2016), p. 23), and this last reference refers to Chen et al., 2009 (Chen, W.; Peng, L.; Wan, N.; Song, L. Mechanism study on the frequent variations of cell-bound microcystins in cyanobacterial blooms in Lake Taihu: Implications for Water Quality Monitoring and Assessments. Chemosphere 2009, 77, 1585–1593.). The correct citation for counting the biological communities of plankton is Chen et al., 2009, should be added in the section 4.1 (lines 274-276). Also, Chen et al. 2009 does not specify which settling chamber is used for phyto or zooplankton (e.g., Utermöhl chamber for phytoplankton??), authors should specify this question in section 4.1 lines (274-276).

However, the quantitative method of counting plankton described in Chen et al. 2009 is carried out with a settling chamber and counting under a microscope, this results in a concentration of cell per mL but the authors express the concentration of plankton taxa in mg/L (Figures 5, 6, 7 and 8)… How is this measurement made and can it be transformed into a concentration in mg/L??

Why not express the concentration of plankton communities in biovolume (mm3/L)?... In fact, the biovolume is the parameter most directly related to cell biomass for algae and cyanobacteria.

Units of concentration of plankton communities are not discussed in section 4.1 either. 

The main results of the work are based on the concentration data of the Microcystis spp. and main zooplankton taxa, if the procedures to quantify the plankton assembalges are not correct they could invalidate the conclusions obtained, and the manuscript must be rejected. Authors must complete and explain this issue in detail.

(iii)PCA and RDA analyzes are not detailed in section 4 `material and methods´, the variables that have been selected to carry out the PCA are not detailed; or if data transformations had to be carried out to make them comparable (e.g., Log10 transformations).

In fact, the Microcystis variables (sum of the species) and each of the Microcystis species are analyzed together through PCA analysis, however, both variables autocorrelate with each other. They should perform the PCA with one or the other variables but not together.

A percentage of explained variability around 50% in the PCA analysis of the global data set could indicate that there are two subsets of data that are explained by different sets of variables???

(iv) In the section 4.2. `Imputation of missing MCs using non-negative latent factor´ the authors comments: `Missing values of intra- 282 or extra-cellular MCs concentrations were imputed using NLF algorithm based on the 283 complete information of environmental factors, cyanobacterial communities (especially 284 Microcystis morphospecies composition), and zooplankton communities.´

I think they should explain how many missing values there are in the data set to assess the statistical validity of the analyzes performed.

Minor questions:

Line 79: Replace `Results and Discussion´ by `Results´

Lines 83-85: The first time a species is cited, the author should also be indicated (i.e., Microcystis aeruginosa, Microcystis wesenbergii, Microcystis novacekii). Also, in the footnote of Figure 1 (lines 94-96) (i.e., M.I; Microcystis ichthyoblabe, M.F; Microcystis flos-aquae, M.V; Microcystis viridis.)

Line 89: The sentence `correlated with MCs quota and concentrations.´ is separated of the previous text.

Line 262: Name of the section 4. `Material and methods´ it's more appropriate than `Experimental section´ because this section also includes statistical methods and analyses, not just experimental methods.

Lines 268-269: The authors mention `Cyanobacterial genera´ in reference to Figure 1. However, in the Figure 1 are mentioned genera and species of cyanobacteria, therefore, it's more appropriate `Cyanobacterial taxa´.

Lines 270-276: `Physiochemical factors used in the building model…´ At what depth were these descriptive limnological parameters taken?... Shan et al. (2019) (section 2.2) cited: `During each transect, water samples were collected at 0.5 m depth of surface layer.´… but in this work some depth was selected?? a mean values of the water column (vertical profile) were made?? or how the authors was proceeded?? (must be specified in section 4.1)

Line 271: `Secchi Disk (SD, in cm)´ this is the name of the instrument, I understand that the measured parameter is the Secchi Disk depth or Secchi depth (must be corrected).

Lines 273-274: `Shaded index´ have other authors proposed it before? it is possible to add a reference on the shaded index that allows to support its utility as a limnological variable. If not, could you explain in material and methods why this index is useful and/or how this conclusion was reached?

Line 284-285 (Section 4.2): `cyanobacterial morphospecies´ should be replaced by `cyanobacterial species´. The authors mentioned `cyanobacterial communities (especially Microcystis morphospecies composition)´ but the names are entities that are currently accepted taxonomically, they are cyanobacterial species (not morphospecies) (footnote of Figure 1 (lines 94-96): M.A; Microcystis aeruginosa, M.W; Microcystis wesenbergii, M.N; Microcystis novacekii, M.I; Microcystis ichthyoblabe, M.F; Microcystis flos-aquae, M.V; Microcystis viridis.), in fact, the authors also mentioned `Cyanobacteria species are in italics´ in the footnote of Figure 1 (lines 94-96).   

[FIGURES AND PLATES]

Table 1: center the values of the first variable (Microcystis) in the 3 columns, and the values of the variable `Extracellular MCs´ in the second column (MLR) (homogenize the number of decimals to 3 like the other columns).

Low quality and low resolution of figures 1, 4, 9.

Figure 1a (Line 90):  In the graph the genera are indicated without `spp.´, except `Anabaena spp.´, it should be homogenized with the others as `Anabaena´.

Also, in the footnote the punctuation mark `;´ between the complete name and the abbreviation name is not correct since it separates two different sentences ((lines 94-96): M.A; Microcystis aeruginosa, M.W; Microcystis wesenbergii, M.N; Microcystis novacekii, M.I; Microcystis ichthyoblabe, M.F; Microcystis flos-aquae, M.V; Microcystis viridis). I suggest other ways to indicate it: e.g., Microcystis aeruginosa = M.A.; Microcystis aeruginosa (M.A.).

Figure 2 (line 108): For consistency with Figure 1, graphs should be indicated with a and b. Figure 2 a (left graph) and b (right graph) should be indicated, indicating it in the footnote. Correct units of Extracellular microcystins (y axis) of the right graph: replace `ug/L´ by `µg/L´.

The coefficient of determination R2 is usually expressed in %  but in this case is a simple linear regression is equivalent to Pearson's coefficient, is a measure of linear dependence between two quantitative variables. However, the authors indicate `P < 0.001´ but it is not indicated in the footnote or in the material and methods section to which it corresponds… What it is `P´??. If it were the p-value of the statistical correlation, it should be indicated in material and methods, and by which statistical test it was obtained (e.g., t-student test).

Figure 3 (lines 111-112): For consistency with Figure 1, graphs should be indicated with a and b. Figure 3 a (up graph) and b (down graph) should be indicated, indicating it in the footnote. Correct units of Extracellular microcystins (y axis) of the down graph: replace `ug/L´ by `µg/L´. Indicate in the graph the variable 'time' or 'months' on the y-axis.

Figure 5 (line 146): Unit letters are overlapped on `(d) Extracellular mycrocistins (µg/L)´. The visualization of the parameters of the y-axis is always better horizontally, here they can be indicated horizontally in two alternating levels.

Figure 6 (line 168): Numbers of the y-axis graphs are overlapped on graduation lines.

Figure 7 (line 173): Numbers of the y-axis graphs are overlapped on graduation lines.

Figure 8 (line 179): Numbers of the y-axis graphs are overlapped on graduation lines. Also, the number 10 (vertical disposition) is overlapped with the x-axis name `Cladocerans (mg/L)´. Correct units of Extracellular microcystins (y-axis): replace `ug/L´ by `µg/L´.

Figure 9 (line 289): Correct the words in the central graphs of the schematic diagram. Replace `varible´ by `variable´, also in the y-axis replace `varibles´ by `variables´. Replace `Imcomplete´ by `Incomplete´.

Supplementary Information

Figure S1 and S2: The authors must include in the `material and methods´ section the interpolation method (e.g., ordinary kriging??) and the R package (or other software) for the realization of the maps of `The distributions of the annual mean of intra/extracellular microcystins (dissolved MCs) across 22 sampling sites in Lake Taihu based on NLF imputation.´

[REFERENCES]

Correct the volume number (issue) format in italics in all references

Correct punctuation mark after year (replace `.´ by  `,´)

Line 426/Reference 37: Replace `1999.´ by `1999,´

Line 433/Reference 41: Replace `2013.´ by `2013,´; also replace `coloniesof´ by `colonies of´

Correct the space in the format of the references

Line 370/Reference 12: Replace `2018,16 (8)´ by `2018, 16(8)´

Line 384/Reference 18: Replace `2020,11 (11)´ by `2020, 11(11)´

Correct the format of the journal abbreviation (check the punctuation marks)

Reference 3: Toxicon. (no punctuation mark) / Toxicon

Reference 9, 13, 14, 45: Harmful. Algae. (no punctuation mark) / Harmful Algae

Reference 18: Atmosphere. (no punctuation mark) / Atmosphere

Reference 19: Inland. Waters. (no punctuation mark) / Inland Waters

Reference 21: Neurocomputing. (no punctuation mark) / Neurocomputing

Reference 30: Plos. One.  (no punctuation mark) / Plos One

Reference 33: Global. Change. Biol. (no punctuation mark) / Global Change Biol.

Reference 34: Microbes. Environ. (no punctuation mark) / Microbes Environ.

Reference 35: Environ Sci Pollut Res (punctuation mark) / Environ. Sci. Pollut. Res.

Reference 37: Aquat Conserv. (punctuation mark) / Aquat. Conserv.  

Reference 38: Science. (no punctuation mark) / Science

Reference 42: Hydrobiologia. (no punctuation mark) / Hydrobiologia

Reference 46: Toxins. (no punctuation mark) / Toxins

Reference 47: IEEE. Access. (no punctuation mark) / IEEE Access

Reviewer 2 Report

The article entitled ‘Revealing physiochemical factors and zooplankton influencing 2 Microcystis bloom toxicity in a large-shallow lake using Bayesian machine learning’ is an original contribution, where the authors gently approach to the very important environmental problem, which is cyanotoxin persistence in water and methods of predicting their occurrence in freshwater habitats. Using Bayesian additive regression tree (BART) the authors demonstrated how Microcystis biomass and microcystin concentration is correlated with physicochemical parameters and zooplankton. Although the problem presented by the authors is relevant, the statistical analyses are appropriate and well described, I have some difficulties with this manuscript.

I am missing stronger background for this study. Why work like this is relevant? Why now? Perhaps there is some environmental problem that makes work like this significant? Furthermore, more than a half of the introduction is devoted to the description of the analytical method. What I think should be appropriate here, is not the method description, but the problem of microcystins persistence in water and challenges, where the method is helpful and applicable. Another problem is, that the study has no clearly stated aim and/or hypothesis. Finally, what I think is a serious deficiency, is a lack of any description or details on methods of microcystin concentration measurement and sampling, identification and biomass calculation of zooplankton. Overall, the study seems to be more a verification if a sophisticated analysis is capable to predict some environmental phenomenon.

Some minor issues:

L33: ‘Cyanotoxin can cause public health…’, should be ‘Cyanotoxins’; please revise

L88: Abbreviations are not explained before in the text, what is making the reader confused

L89: Both Figure 1 and 4 plots are of terrible quality

Section 2: ‘Results and Discussion’ is actually Results section only. I do not see the point why the authors entitled the section like this.

Reviewer 3 Report

The manuscript is dedicated to describing the NLF-BART model, a hybrid machine learning model that can be used to evaluate relationships between MC dynamics and their key factors. Recently, many machine learning approaches have been used to identify correlations and assess  the drivers of complex ecological dynamics. This is a very promising scientific direction.

The BART method showed good results in assessing the dynamics of cyanotoxins in the studied Lake Taihu. The application of Bayesian ensemble modeling methodology to assess the risk of harmful algal blooms can be a very good approach in a case of incomplete dataset.

This article shows the perspectives of the BART method.

The paper is interesting to me, I would suggest it for publication.

Author Response

"Please see the attachment

Round 2

Reviewer 1 Report

Dear Editor,

I recommend that the manuscript be accepted for publication in the present form. I have carefully reviewed the manuscript and the authors have made the appropriate changes to clarify the experimental procedures and the results obtained, the authors have improved the manuscript considerably (especially the section of `Material and Methods´). The manuscript is well written and the narrative is very fluid with a good level of English. The discussion is consistent with the results obtained and the specialized literature on the subject.

Reviewer 2 Report

The authors have done a good job in revising the paper. I think it can be published in present form.